# Evaluation of the Role of AID-Induced Mutagenesis in Resistance to B-Cell Receptor Pathway Inhibitors in Chronic Lymphocytic Leukemia

**DOI:** 10.3390/cimb47121031

**Published:** 2025-12-10

**Authors:** Chiara Pighi, Alessandro Gasparetto, Elisa Genuardi, Jianli Tao, Qi Wang, Candida Vitale, Valentina Griggio, Rocco Piazza, Sabino Ciavarella, Marta Coscia, Simone Ferrero, Alberto Zamò, Claudia Voena, Roberto Chiarle

**Affiliations:** 1Department of Molecular Biotechnology and Health Sciences, University of Turin, 10126 Turin, Italyalessandro.gasparetto@unito.it (A.G.);; 2IRCCS Sacro Cuore Don Calabria Hospital, Department of Pathology, Negrar di Valpolicella, 37024 Verona, Italy; 3Department of Pathology, Children’s Hospital Boston, Harvard Medical School, Boston, MA 02115, USA; 4Division of Hematology, A.O.U. Città della Salute e della Scienza di Torino, 10126 Turin, Italy; 5Department of Medicine and Surgery, University of Milano-Bicocca, 20126 Milan, Italy; 6Hematology Unit, Oncology Department, IRCCS Istituto Tumori “Giovanni Paolo II”, 70124 Bari, Italy; 7Hematology Division, ASST Sette Laghi, Ospedale di Circolo, 21100 Varese, Italy; 8Department of Medicine and Surgery, University of Insubria, 21100 Varese, Italy; 9Institute of Pathology, University of Würzburg, 97080 Würzburg, Germany

**Keywords:** Chronic Lymphocytic Leukemia (CLL), activation-induced cytidine deaminase (AID), idelalisib, ibrutinib, drug resistance

## Abstract

Chronic lymphocytic leukemia (CLL) is the most common leukemia in Western countries, and B-cell receptor (BCR) pathway inhibitors such as idelalisib and ibrutinib are currently established therapies for CLL. Although effective, these drugs frequently lead to resistance, but the mechanisms are still not fully understood. Activation-induced cytidine deaminase (AID) is a B-cell enzyme essential for antibody diversification. However, it can also introduce off-target mutations, leading to genomic instability. This study investigates whether treatment with BCR pathway inhibitors increases AID activity in CLL and whether this activity contributes to the development of drug resistance. Peripheral blood samples from CLL patients were collected before and after treatment with idelalisib or ibrutinib. Targeted sequencing was used to identify mutations in known AID off-target genes. Concurrently, AID-wild type (AID-WT) and AID-knockout (AID-KO) CLL cell lines were established and subsequently exposed to escalating doses of BCR pathway inhibitors to develop drug-resistant models. In patient samples, treatment with BCR pathway inhibitors was associated with an increase in AID-dependent mutations in off-target genes, including *BCL2*, *MYC*, and *IRF8*. The in vitro models efficiently recapitulated the patients’ data, as only AID-WT CLL cells accumulated mutations in the same AID off-target genes after drug exposure. However, no mutations were detected in genes that could mediate drug resistance. We conclude that BCR pathway inhibitors enhance AID mutational activity in CLL, but this does not appear to be directly involved in driving drug resistance. AID-targeted loci may nonetheless serve as biomarkers for monitoring genomic instability during treatment and inform further study.

## 1. Introduction

Typically occurring in adulthood, chronic lymphocytic leukemia (CLL) is the most prevalent form of leukemia in Western countries. It is defined by the monoclonal growth of CD5+ B-lymphocytes, with an 88% 5-year survival rate [1]. CLL cells exploit B-cell receptor (BCR) signaling for proliferation, and its inhibition is crucial for disease treatment.

Ibrutinib and idelalisib are two first-in-class BCR pathway inhibitors, approved for the treatment of patients with CLL. Ibrutinib, a covalent inhibitor of Bruton’s tyrosine kinase (BTK), is approved both as first-line therapy and for relapsed or refractory CLL, including patients harboring *TP53* mutations or 17p deletions, who typically exhibit poor responses to chemoimmunotherapy. Idelalisib, a selective inhibitor of the phosphatidylinositol 3-kinase (PI3K) δ isoform, is approved in combination with rituximab for patients with relapsed CLL who are not suitable for cytotoxic therapy [2]. Although both agents have markedly improved progression-free and overall survival, resistance to BCR pathway inhibition inevitably arises. Such resistance frequently involves mutations in the drug target [1], including *BTK* (notably at codons 316 and 481), or activating mutations in *PLCG2* that restore downstream BCR signaling despite BTK inhibition [3]. For example, the *BTK* C481S mutation abolishes covalent drug binding and restores BCR signaling. Additional BTK mutants, including kinase-impaired mutations such as L528W and gatekeeper mutations like T474I, have been increasingly recognized across different covalent and non-covalent BTK inhibitors [4]. Likewise, gain-of-function mutations in *PLCG2* (e.g., R665, S707) can bypass BTK inhibition through constitutive PI3K activation [4]. Importantly, resistance mechanisms extend beyond the BCR axis and include alterations in EGR2, NF-κB, PTEN/AKT, MAPK [5] or TLR-driven bypass pathways, and a protective tumor microenvironment [3,6,7]. Overexpression of MDR1/P-glycoprotein has also been implicated as an alternative resistance route, particularly in the context of PI3Kδ inhibition, enabling increased efflux of targeted agents [8]. The origin of these mutations and the cellular mechanisms sustaining acquired resistance remain not fully understood.

The APOBEC family member Activation-Induced cytidine Deaminase (AID) is an enzyme expressed specifically in B-cells. Its primary targets include immunoglobulin genes, initiating critical processes such as class switch recombination and somatic hypermutation [9]. AID induces canonical and non-canonical mutational signatures in CLL [10]. Canonical activity targets cytidines within WRCY motifs, generating C to T or C to G substitutions at immunoglobulin loci through base-excision repair. Non-canonical activity arises from off-target deamination events and error-prone repair by mismatch or translation polymerases [11]. Increased AID activity amplifies genomic instability by affecting non-immunoglobulin genes, including the initiation of oncogenic mutations and the formation of chromosomal translocations linked to blood cancer development [12,13]. AID has been shown to introduce off-target mutations in multiple proto-oncogenes, including *MYC*, *PAX5*, *BCL6*, and *PIM1* [14]. AID is expressed not only in germinal center-derived lymphomas but also in CLL and several other mature B-cell malignancies [15,16]. Experimental models demonstrate that aberrant AID activity can accelerate B-cell lymphomagenesis by promoting chromosomal translocations, secondary oncogenic alterations, and the accumulation of widespread non-Ig mutations [17]. For example, in *Emu-Myc* transgenic mice, AID facilitates the acquisition of additional driver mutations such as those in *Pim1*, supporting its role in clonal evolution [18].

The PI3Kδ pathway normally inhibits AID activity. We have shown that BCR pathway inhibitors, such as idelalisib and ibrutinib, function by blocking PI3Kδ and thereby increasing genomic instability in B cells, including in CLL models [9,10]. (Figure 1A). These findings suggest that AID-mediated mutations may contribute to CLL disease evolution, progression, and potentially drug resistance. AID activity correlates with drug resistance in other blood cancers, such as Chronic Myeloid Leukemia [19] and Burkitt lymphoma [20]. In solid tumors, APOBEC3A-induced genomic instability is recognized as a key player in lung cancer drug resistance [21].

To this date, the role of AID-induced mutations has not been definitively evaluated in the context of resistance to ibrutinib or idelalisib, and the interplay between BCR pathway blockade, AID activity, and drug resistance remains largely unexplored. Here, we experimentally investigated the presence of AID-dependent mutations accumulated during treatment in a cohort of patients with CLL, and then, using an in vitro model, we evaluated the possible role of AID in the development of resistance to BCR pathway inhibitors in CLL.

## 2. Materials and Methods

### 2.1. Samples from CLL Patients

DNA was extracted from peripheral blood samples obtained from patients with CLL before and after therapy (untreated *n* = 10; idelalisib *n* = 8; ibrutinib *n* = 11; for a total of 58 biological samples corresponding to pre- and post-treatment timepoints). All patients were taken in charge at “Città della Salute e della Scienza” hospital in Turin, Italy. All patients provided written informed consent. All cases were diagnosed according to the international guidelines and consented according to internal protocols. Cohort baseline information is shown in Appendix A. Extracted genomic DNA was used for targeted sequencing of AID off-target genes to quantify treatment-associated mutation accumulation.

### 2.2. Cell Lines and Reagents

The human leukemia MEC1 cell line (CVCL_1870; Chronic Lymphocytic Leukemia) was purchased from DSMZ (Leibniz, Germany) and was cultured in RPMI 1640 medium (Pan Biotech, Aidenbach, Germany) supplemented with 10% fetal bovine serum (FBS), penicillin-streptomycin (100 units per ml) and L-glutamine (2 mM). Cell lines were tested negative for mycoplasma contamination. Idelalisib (CAL-101, GS-1101; PI3Kδ inhibitor) and ibrutinib (inhibitor of Bruton’s tyrosine kinase) were purchased from Selleckchem (Houston, TX, USA).

### 2.3. Generation of AID-Knockout Cell Line Clones

HEK293FT cells (CVCL_6911) purchased from Invitrogen (Carlsbad, CA, USA) were cultured in DMEM containing 10% FBS. For lentiviral particle production, 5.5 million HEK-293FT cells were seeded into 10 cm dishes. The next day, these cells underwent transfection using the calcium phosphate method, employing 7.2 μg of the lentiCRISPR v2 plasmid, 3.6 μg of VSVG, 3.6 μg of RSV-REV, and 3.6 μg of PMDLg/pPRE. Eight hours post-transfection, the culture medium was changed. After 36 h, the viral supernatant was collected, filtered through a 0.45 μm filter, pooled, and used fresh or stored frozen. A single guide RNA (sgRNA-CCTCCGCTACATCTCGGACTGG) was designed to target exon 3 of the *AICDA* gene using a CRISPR design tool available at https://portals.broadinstitute.org/gppx/crispick/public (10 June 2019). For transducing MEC1 cells with CRISPR/Cas9 lentiviruses targeting AID-exon3, a total of 400,000 human neoplastic cells were plated into 6-well plates at a concentration of 200,000 cells per ml. Lentiviral transduction involved adding lentiviral supernatant and spinning it for 1.5 h at 2400 r.p.m. in the presence of polybrene (6 μg/mL). After 48 h post-transduction, cells were subjected to selection with puromycin at a concentration of 0.2 μg/mL for 3 days. The selected cells were then individually seeded into 96-well plates via serial dilutions. After 3–4 weeks of culture, cells derived from each colony were used for evaluating AID-knockout by Western blotting and the genomic DNA was used for targeted DNA sequencing of the sgRNA target region. The genomic region surrounding the CRISPR target sites was analyzed using Whole Exome Sequencing (WES) data.

### 2.4. Generation of Resistant Clones

AID-KO and AID-WT MEC-1-derived clones were subjected to chronic treatment with idelalisib or ibrutinib to induce drug resistance. Cells were initially plated, and over time, drug concentrations were incrementally increased in the culture medium, starting from 1/10 of the basal IC50 value calculated for the parental MEC-1 cell line. Specifically, AID-WT clones (AID.WT1, AID.WT2) and AID-KO MEC-1 cell lines (AID.KO1, AID.KO2) were treated with an initial dose of 50 nM of idelalisib or 200 nM of ibrutinib, and this treatment was sustained for a minimum of 6–9 months.

### 2.5. Cell Viability Assay

The cell viability assays were performed using CellTiter-Glo^®^ (Promega, Madison, WI, USA), following the manufacturer’s instructions. In summary, cells were plated in a 96-well white plate with their respective medium and drugs. After 72 h, the CellTiter-Glo^®^ reagent was added to each well at a 1:1 ratio, and luminescence was then measured using the GloMax^®^-Multi Detection System (Promega, Madison, WI, USA) at a wavelength of 550–570 nm. The drugs were diluted in DMSO. The results were normalized by subtracting the background signal from each well, and the dose–response relationship was obtained using GraphPad Prism 9.2.0 software.

### 2.6. IC50 Analysis

The cell viability data were used to generate pharmacological sensitivity curves. Specifically, each experiment was repeated at least 3 times. The curves were calculated using GraphPad Prism 9.2.0 software, employing a 4-parameter logistic interpolation model, and graphed using a nonlinear sigmoidal dose–response regression model. Each point on the graph (mean ± SEM) represents the ratio of cell viability for the treated cells compared to the untreated control.

### 2.7. Idelalisib and Ibrutinib Treatment

Cells were diluted in RPMI 1640 culture medium to a density of 500,000 cells/mL and allowed to grow for 16 h. Once they reached the exponential growth phase, a varying drug concentration or a volume of DMSO corresponding to the maximum drug dose used was added to the culture medium as a negative control. After 8 h of treatment, cells were harvested and washed with cold phosphate-buffered saline (PBS). Cell pellets were frozen and stored at −80 °C until subsequent Western blot analysis.

### 2.8. Western Blot Analysis

Protein extraction and Western Blot analysis were conducted as follows: whole-cell extracts were prepared from cellular pellets. Protein extraction was performed using GST-FISH buffer containing 10 mM MgCl2, 150 mM NaCl, 1% NP-40, 2% Glycerol, 1 mM EDTA, and 25 mM HEPES (pH 7.5), supplemented with protease inhibitors (Roche), 1 mM phenylmethanesulfonylfluoride (PMSF), 10 mM NaF, and 1 mM Na_3_VO_4_. Protein extracts were then clarified by centrifugation at 12,000 r.p.m. for 15 min, and the resulting supernatants were collected. Protein concentration was determined using the Bio-Rad protein assay method. Subsequently, 30 μg of protein samples were loaded onto 10% Mini-PROTEIN TGX gels (BIO-RAD, Waltham, MA, USA), followed by transfer (Trans-Blot Turbo Transfer System; BIO-RAD, Waltham, MA, USA) onto nitrocellulose membranes (BIO-RAD, Waltham, MA, USA). The membranes were then blocked with 5% non-fat milk (BIO-RAD, Waltham, MA, USA) for 1 h. Primary antibodies used for immunoblotting included the following: mouse anti-human-AID (1:500, ZA001, Life Technologies, Carlsbad, CA, USA), mouse anti-β-actin (1:3000, 8H10D10, Cell Signaling Technology, Danvers, MA, USA), rabbit monoclonal anti-phospho-AKT (S473) (1:500, D9E, Cell Signaling Technology, Danvers, MA, USA), rabbit monoclonal anti-AKT (pan) (1:1000, #2920, Cell Signaling Technology, Danvers, MA, USA), rabbit anti-phospho-BTK (Y223) (1:1000, D9T6H, Cell Signaling Technology, Danvers, MA, USA), rabbit anti-BTK (1:1000, D3H5, Cell Signaling Technology, Danvers, MA, USA), rabbit anti-phospho-ERK1/2 (Y202/204) (1:1000, #9101, Cell Signaling Technology, Danvers, MA, USA), rabbit anti-ERK1/2 (1:1000, #9102, Cell Signaling Technology, Danvers, MA, USA). Membranes were developed using an ECL solution (GE Healthcare, Chicago, IL, USA) and Chemidoc Imaging System (BIO-RAD, Waltham, MA, USA).

### 2.9. Targeted DNA Sequencing

Genomic DNA was extracted from the peripheral blood cells of CLL patients using the MagCore automated extraction following the manufacturer’s protocol. The quality and quantity of the extracted DNA were verified using NanoDrop and Qubit (Thermo Fisher Scientific, Waltham, MA, USA). A targeted genomic region of 1 Kb upstream of the transcription start site (TSS) of 23 off-target genes of AID and 4 control genes was sequenced using a custom panel designed with Illumina’s AmpliSeq technology. The genes included in the panel are *MYC exon 1*, *MYC exon 2*, *RhoH*/*TTF*, *PAX5*, *PAX5-SE*, *PIM1*, *BCL6*, *TCL1A*, *CXCR4*, *IRF4*, *BCL2*, *MIR142*, *BTG2*, *LRMP*, *BCL7A*, *BACH2*, *SOCS1*, *IRF8*, *S1PR2*, *BIRC3*, *CD74*, *CD83*, *MS4A1*, *BTK*, *ICAM*, *TFRC*, *LCP1*, *MYBL1*. In particular, 10 ng of genomic DNA was used for each primer pool to amplify the 138 amplicons of the panel. The libraries were assessed and quantified using the Agilent D1000 High Sensitivity kit on the Tape Station (Agilent Technologies, Lexington, MA, USA) and then mixed at equimolar concentrations before being sequenced on the Illumina Miseq platform (Illumina, San Diego, CA, USA). Sequences with a quality score lower than 20 or a length less than 50 were removed. Samples with a read count lower than 100 were excluded from the analysis. The remaining sequences were used for assessing mutational frequency. The obtained sequences were aligned to the reference sequence using BLASTN (MIT, version 0.2.3.9000). Mutations underwent several selection steps. Initially, they were evaluated using Neighbourhood Quality Standard criteria, selecting only those with a Phred score of 30 and achieving a score of 20 in the 5 adjacent bases. Mutations within 5 bases of more than two other mutations were excluded. Mutations within 2 bases of insertion and deletion sites were also excluded. Additionally, bases with a mutational frequency >0.01 were excluded as they were SNP-dependent, and those with >0.02 were classified as SNPs. Finally, the mutational frequency was calculated using both the forward and reverse sequences if possible. To calculate the average base mutations, only C>T and G>A transitions were evaluated. To test whether AID-dependent mutations correlate with specific clinical/biological features, the total AID-dependent mutations accumulated upon treatment (change from the pre-treatment sample) of a specific subgroup (e.g., patients with *TP53* mutated) were calculated. Significance was calculated with the one-way ANOVA test with Tukey’s correction for multiple comparisons.

### 2.10. Whole Exome Sequencing (WES)

Raw Fastq data were initially quality tested using FastQC (https://www.bioinformatics.babraham.ac.uk/projects/fastqc/, accessed on 4 October 2021). Paired fastq reads were then aligned with BWA v.0.7.12 using human GRCh38/hg38 as the reference genome. Aligned reads in bam format were sorted and indexed using Samtools v.1.1. Single nucleotides were assessed using Strelka2 v.2.9.10 with both germline and somatic configurations using the -exome flag. Variants in vcf format were further annotated using CEQer. Structural variants were analyzed using Manta v.1.6.0 and copy number abnormalities using CopywriteR v.2.22.0 by applying a bin size of 20Kb and CEQer. Alignments were manually inspected using the Integrative Genomics Viewer. Analysis of DNA sequence contexts of somatic mutations was performed with Mutalisk.

## 3. Results

### 3.1. BCR Inhibition Increases AID-Mediated Mutations in Patients with CLL

We previously demonstrated that treatment with idelalisib and ibrutinib results in elevated AID expression in activated B lymphocytes, thereby contributing to increased genomic instability [17] (Figure 1A). Despite a pronounced clinical response, CLL patients treated with BCR pathway inhibitors frequently relapse. To verify the contribution of enhanced AID activity in the acquisition of increased mutational rate, we analyzed AID-mediated genomic mutations in blood samples from a cohort of patients with CLL (Figure 1B) treated with idelalisib (*n* = 8) or ibrutinib (*n* = 11) for about 12 months and a control cohort (*n* = 10) comprising patients with CLL who were not exposed to BCR pathway inhibitors but received standard-of-care chemotherapy. The frequency of AID-dependent mutations accumulated during BCR-inhibitor treatment was assessed by targeted sequencing covering up to 1000 bps upstream of the transcription starting site (TSS) of a panel of 27 genes (23 AID off-target genes; 4 non-target control genes—Appendix A) in paired pre- and post-treatment CLL samples. As targets, we selected a list of genes or genomic regions known to be frequently hypermutated in CLL and B-cell lymphomas due to AID-mediated deamination. An increase in AID-dependent mutations was not detected in untreated CLL patients. In contrast, a significant increase in the AID-mediated mutational rate was observed in *BCL2*, *CD83*, *MYC exon1*, *IRF8*, *PAX5-SE*, *PIM1*, *RHOH*/*TTF*, and *TCL1A* genes during idelalisib treatment and in *BIRC3*, *BTG2*, and *LRMP* genes during ibrutinib treatment (Figure 1C). No changes in mutation rates were detected in any of the four control genes (non-AID target genes) sequenced in every patient included in the study (Appendix A).

To test whether patients enrolled in our study accumulated BTK mutations after ibrutinib treatment, we performed targeted sequencing of BTK, focusing on codons 316 and 481, two residues frequently mutated in patients with CLL who had become resistant to ibrutinib treatment [22]. Notably, none of the patients in this study acquired these mutations (Figure 1B) despite an increased AID-dependent mutational rate, consistent with the low BTK mutation frequency in the CLL setting after ibrutinib treatment (3% in stable disease and 30% in progression, respectively [23]).

In addition, we did not find any correlation between AID-mutational signatures and *TP53* mutations, immunoglobulin hypermutation status (IGHV status), or disease progression as per relevant clinical features in patient clinical reports (Figure 1B).

Thus, we demonstrate that treatment with BCR pathway inhibitors clearly enhances AID-dependent mutagenesis in specific off-target genes, and this increased mutational activity does not correlate with CLL-relevant clinical features.

### 3.2. Generation of AID-Knockout CLL Models to Study Resistance to BCR Inhibitors

To experimentally examine the role of AID in developing resistance to idelalisib and ibrutinib, we used CRISPR-Cas9 technology to knock out AID in a human CLL cell line, MEC-1 [24], a human cell line derived from prolymphocytic CLL and previously used as a powerful model to detect AID-dependent genome instability during idelalisib treatment [13,17]. We designed a single guide RNA (sgRNA) targeting exon 3 of the *AICDA* gene to generate a frame-shift deletion of the AID-encoding gene (Appendix A). We characterized several clones by Western Blot analysis of the AID protein and then validated them by sequencing. We obtained multiple MEC-1 clones that were either wild type (AID.WT1 (10_6), AID.WT2 (10_8)) or knock out (AID.KO1 (10_3.19), AID.KO2 (10_3.28)) for AID expression (Appendix A). For further confirmation, we sequenced exon 3 of the AICDA gene in the AID-KO clones and found that clones AID.KO1 and AID.KO2 exhibited a near-complete deletion of 5 nucleotides and a partial deletion of 13 nucleotides, respectively, at the Cas9 cleavage site region. These alterations resulted in the depletion of AID expression. (Appendix A). A pair of clones for both WT and KO conditions was selected to reduce the clonal bias.

Next, we investigated whether AID mutagenic activity contributes to drug resistance by subjecting AID WT and AID KO clones to the selective pressure of BCR pathway inhibitors, aiming to model in vitro what is observed in patients with CLL. The selected cell clones were made resistant to BCR pathway inhibitors by exposure to increasing concentrations of the drugs in culture medium, starting at approximately one-tenth of the 50% dose-inhibition (IC50) calculated in the MEC-1 cell line. Thus, we generated idelalisib-resistant and ibrutinib-resistant cellular models for each of the AID WT and KO clones (Appendix A). As shown in Figure 2A and Appendix A, resistant cell lines exhibited a significantly higher IC50 than their parental (sensitive) counterparts, confirming their resistance to therapeutic doses of the respective inhibitors.

These models provided a platform to investigate the role of AID in acquired resistance to BCR inhibitors in CLL.

### 3.3. AID-KO Does Not Prevent the Emergence of Resistance to BCR Pathway Inhibitors

To study the signaling in resistant clones, sensitive and resistant cell lines (MEC-1 AID.WT1, AID.WT2, AID.KO1, and AID.KO2) were cultured in the presence of different concentrations (0, 1, 5, 10, 20 μM) of the respective drug (idelalisib or ibrutinib) for 8 h. Subsequently, cells were harvested, and pathway activation was assessed by Western Blot analysis (Figure 2B). AID expression was increased upon treatment, as we previously demonstrated (Figure 2B) [13,17]. Both AID WT or KO-resistant clones showed a markedly reduced blockade of AKT phosphorylation by idelalisib or ibrutinib compared to AID WT or KO-sensitive clones (Figure 2B). The resistant MEC-1 AID.WT2 clone also showed a reduced blockade of ERK phosphorylation compared to the corresponding sensitive clone (Figure 2B). The inability to block AKT and/or ERK phosphorylation is one key feature of resistance to idelalisib or ibrutinib, as reported in patients. Our data demonstrate that AID WT or KO cells can recapitulate the essential signaling features of the mechanism underlying BCR pathway inhibitor resistance in vitro [5,25]. Notably, the signaling mechanisms of resistance were similar in AID WT or KO clones, suggesting that AID expression did not correlate with a specific resistance mechanism.

Furthermore, CLL cell lines were genetically characterized by sequencing the same gene panel as in the patient cohort. Overall, idelalisib-resistant clones accumulated more AID-dependent mutations in the analyzed genes than their ibrutinib-resistant counterparts, which may be partly due to the higher level of resistance achieved by the AID-WT clones under idelalisib (Appendix A). Analyzing gene by gene, idelalisib-resistant AID-WT clones exhibited significantly increased mutation rates in *BCL2*, *MYC exon1*, *CXCR4*, and *PIM1* genes, while AID-WT ibrutinib-resistant clones showed a considerably higher mutation frequency at the *IRF4* locus (Figure 2C). As expected, the AID-KO clones exhibited no increase in AID-dependent mutations in any of these AID-off target genes (Figure 2C) or the selected non-target genes (Appendix A). Thus, our in vitro model recapitulated the AID mutational activity observed in patients treated with BCR pathway inhibitors, as chronic exposure to idelalisib or ibrutinib led to an accumulation of AID-dependent mutations only in AID-WT cells, not in AID-KO B-cells.

Lastly, we asked whether the AID mutagenic activity could genetically explain the development of resistance to BCR pathway inhibitors. Despite their low frequency, activating mutations in genes involved in the MAPK/ERK pathway (such as *MAP2K1*, *BRAF*, *ERBB4*, and *KRAS*) are reported to be responsible for drug resistance in patients [5,26]. Thus, we searched for mutations in these pathways using whole-exome sequencing (WES) in AID WT or KO cell clones. Our study focused on the most common pathways associated with CLL drug resistance [5,26] comparing copy-number variations and single-nucleotide mutations in the exome before and after the development of idelalisib resistance. This analysis did not identify any clinically significant alterations accumulating during drug treatment (Appendix A).

Thus, AID promotes mutagenesis in CLL cells under BCR inhibition, suggesting that the mutagenic activity of AID does not contribute to the emergence of resistance to BCR pathway inhibitors and that resistance likely arises through adaptation rather than mutations in other driver oncogenes.

## 4. Discussion

In this study, we explored whether AID activity is increased in CLL patients treated with BCR pathway inhibitors and whether this increase plays a role in drug resistance. Previous work has shown that PI3Kδ blockade promotes AID expression and genomic instability in normal and malignant B cells [13,17], providing a mechanistic rationale for the hypothesis that therapeutic inhibition of PI3Kδ or BTK may create a permissive condition for off-target AID activity. Consistent with this premise, we found that patients with CLL receiving BCR pathway inhibitors for approximately one year accumulate mutations in AID off-target genes, and this was absent in controls. These findings extend earlier reports of AID-driven mutagenesis in CLL and other B-cell malignancies and support the notion that BCR pathway inhibition may induce genomic stress. The increased mutational burden observed in genes such as *BCL2*, *CD83*, *MYC exon1*, *IRF8*, *PAX5-SE*, *PIM1*, *RHOH*/*TTF*, and *TCL1A* during idelalisib treatment and *BIRC3*, *BTG2*, *LRMP* during ibrutinib treatment mirrors the known spectrum of AID activity described in CLL and diffuse large B-cell lymphoma, where AID is responsible for widespread, off-target deamination events that contribute to tumor evolution [27]. Notably, the accumulation of mutations in our patient cohort did not correlate with *TP53* status, IGHV mutational profile, clinical progression, or the emergence of BTK mutations commonly associated with ibrutinib resistance [4]. This observation is in line with genomic analyses demonstrating that while AID can generate numerous low-frequency lesions, only a minority of these events become fixed and pathogenically relevant.

Furthermore, we were able to recapitulate this characteristic AID-mediated mutational activity in experimental models of AID WT or KO CLL lines, mechanistically corroborating the findings in patients. We reproduced the AID-mediated mutational signatures observed in patients and demonstrated that AID is responsible for the accumulation of off-target mutations during prolonged drug exposure. Despite these differences in mutational load, both AID-WT and AID-KO clones acquired resistance to idelalisib or ibrutinib with similar efficiency. Resistant cells from both genotypes exhibited diminished inhibition of AKT and ERK phosphorylation, key hallmarks of resistance observed clinically [5,26], indicating that AID activity, while contributing to genomic instability, is dispensable for the development of resistance. Whole-exome sequencing did not reveal clinically actionable alterations in canonical resistance pathways (MAPK/ERK or PI3K/AKT), reinforcing the concept that resistance in these models likely arises from adaptive, non-genetic mechanisms.

These results fit within an increasingly recognized framework in which resistance to BCR pathway inhibitors is multifactorial and often independent of point mutations [23]. Several alternative mechanisms have been implicated in idelalisib and ibrutinib resistance, including activation of EGR2, NF-κB, and pro-survival signaling pathways, rewiring of PI3K/AKT and MAPK cascades, TLR-mediated bypass of the BCR signal, microenvironmental support, dysregulated CXCR4 signaling, and transcriptional and epigenetic adaptation that restore downstream survival pathways even in the absence of upstream BCR activity [5,6,7,26,28]. Our data are consistent with these models: resistance arises in AID-WT and AID-KO cells despite the absence of acquired mutations in BTK, PI3Kδ, or MAPK/ERK oncogenic drivers, suggesting that non-genetic alterations rather than exome mutations may be the predominant route to resistance in this context.

However, our study is not free from limitations: the number of cell lines utilized, the analysis limited to the exome, and the relatively small clinical cohort allow only a partial understanding of the phenomenon. Further investigation would be required to determine whether AID activity plays a significant role in resistance, taking into consideration, for example, the possible AID involvement in super-enhancer hypermutation, as observed in the Diffuse Large B-cell Lymphoma setting [27], or large genomic aberrations observed during CLL transformation to Richter syndrome.

Lastly, our results show that PI3Kδ and BTK inhibition increases genomic instability, suggesting that AID activation could serve as a biomarker of treatment-induced mutagenic stress. Rather than predicting resistance, such a biomarker could identify patients experiencing rapid genomic diversification, a state linked to clonal evolution and potential relapse. Instability at AID target genes may thus allow early monitoring of genomic remodeling and the identification of patients at risk of long-term progression, even in the absence of canonical resistance mutations.

## 5. Conclusions

We demonstrated that pharmacologic inhibition of the BCR pathway in patients with CLL induces off-target genomic mutagenesis mediated by AID, expanding prior observations of PI3Kδ-blockade-induced genomic instability [17]. We further validated these results using an in vitro model of CLL. However, cellular models did not demonstrate a causal link between AID activity and the emergence of resistance to idelalisib or ibrutinib. Both AID-WT and AID-KO cells developed resistance with similar kinetics and signaling phenotypes, suggesting that AID-driven mutations are not necessary for resistance acquisition. Our findings highlight the need for larger, longitudinal genomic studies to define the temporal dynamics of AID activation during therapy and to determine whether sustained AID-mediated genomic remodeling contributes to disease evolution or relapse over extended treatment intervals. Future research should integrate multi-modal approaches to elucidate the full spectrum of resistance mechanisms, including the MAPK pathway [5] and the tumor microenvironment [29]. Additionally, given that BCR pathway inhibition increases instability at AID target genes, these loci may serve as biomarkers of therapy-induced mutagenic stress, offering a strategy to detect early genomic remodeling and anticipate acquired resistance. Clarifying how this instability shapes clonal evolution will be essential for refining therapeutic schedules and developing strategies that mitigate long-term disease progression under targeted therapy.

## Figures and Tables

**Figure 1 cimb-47-01031-f001:**
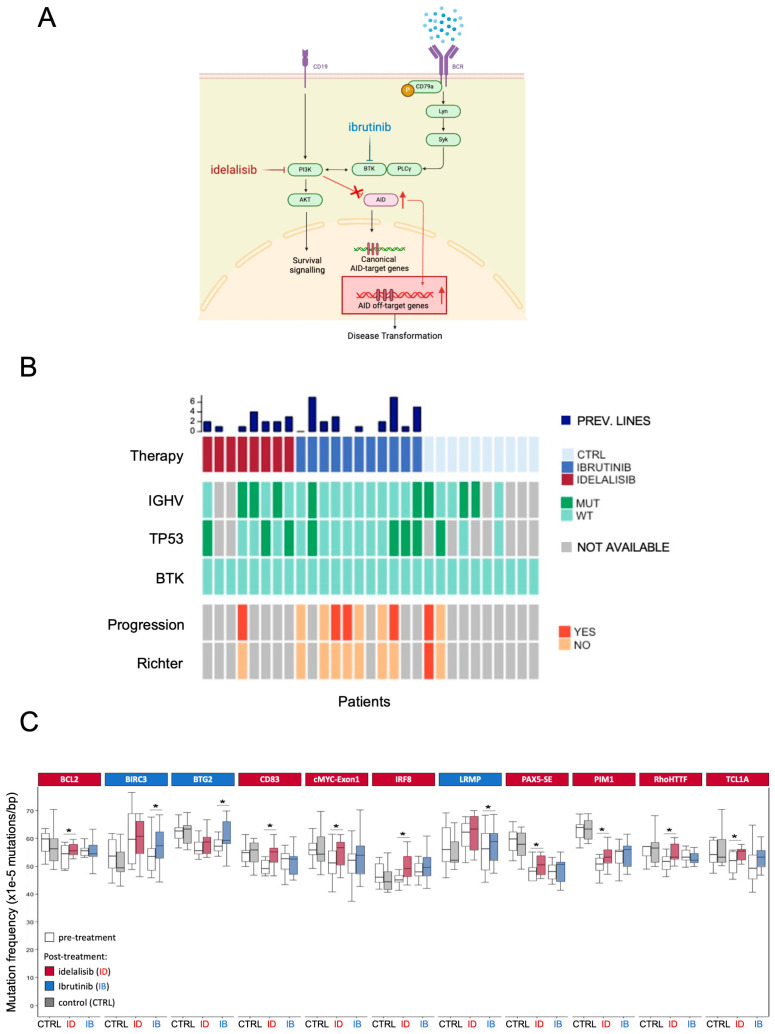
(**A**) Schematic representation of the interaction between BCR-pathway inhibition and AID expression. After idelalisib (red) or ibrutinib (blue) treatment, AID expression rises, leading to increased AID off-targeting activity. (**B**) The heatmap summarizes the characteristics of the participants in the study. A cohort of 29 patients with CLL was selected: 10 BCR pathway inhibitors not-treated patients, 8 idelalisib-treated, and 11 ibrutinib-treated. Main clinical and genomic characteristics were collected from the clinical records. CTRL = control group; MUT = mutated; WT = wild-type; prev. lines: previous lines of therapy (**C**) Eleven AID off-target genes show a higher AID-dependent mutation frequency after idelalisib (ID—red) or ibrutinib (IB—blue) treatment. The graph indicates only the genes that experienced a significant increase in mutational rate following therapy. Boxplots indicate cumulative frequencies of C>T or G>A transition mutation in DNA samples collected before (pre) and after (post) treatment in each patient (control *n* = 10, idelalisib *n* = 8, ibrutinib *n* = 11). Whiskers extend to a maximum of 1.5× interquartile range beyond the box. *p*-values calculated by paired samples two-tailed Student’s *t*-test (* *p* < 0.05).

**Figure 2 cimb-47-01031-f002:**
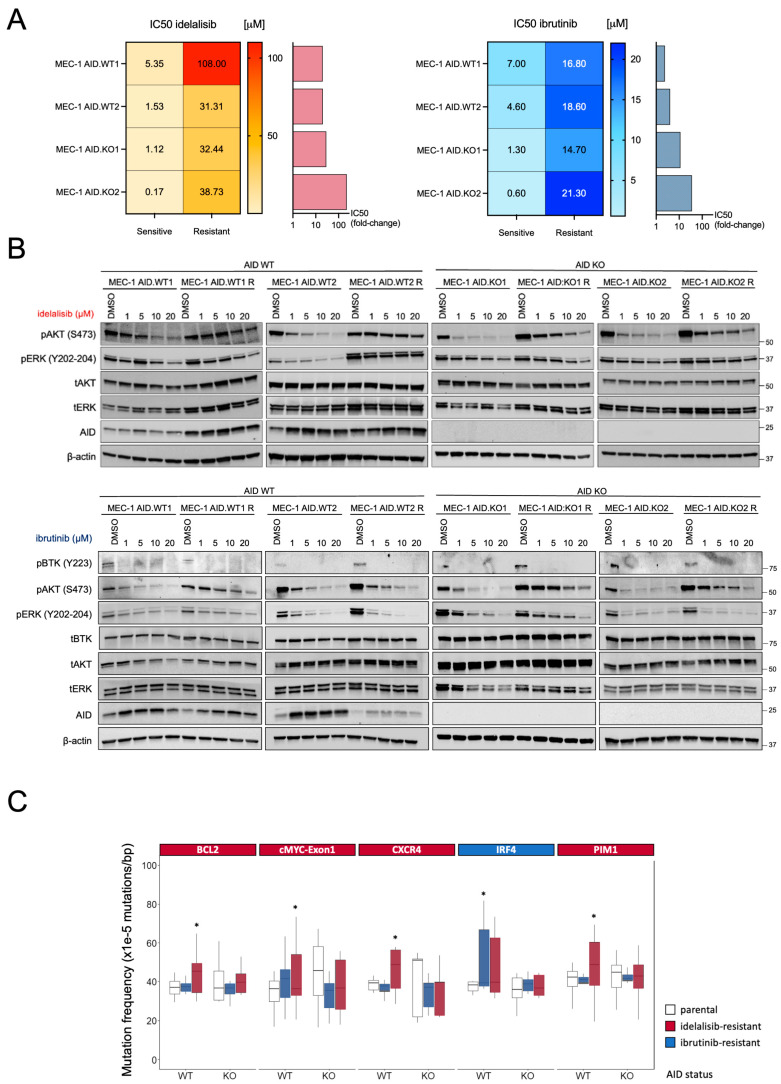
(**A**) The heatmap shows the Inhibiting Concentration 50 (IC50) to idelalisib (orange-red, left) and ibrutinib (light blue-blue, right) of each cell line before and after resistance acquisition. The mean of at least 3 different biological replicates is shown. Bar graphs show the fold-change increase in IC50 for each drug achieved by each clone, compared to the parental counterpart. (**B**) Resistant and parental MEC-1 cells (AID-WT and AID-KO) were treated with either idelalisib (upper) or ibrutinib (lower) with doses up to 20 μM for 8h. Representative Western blot images from at least two independent experiments are shown. (**C**) Mutation frequency of AID off-target genes in sensitive and resistant MEC-1 cell lines. The graph indicates only the genes that experienced a significant increase in mutational rates after the induction of drug resistance. Boxplots indicate cumulative frequencies of C>T or G>A mutations in MEC-1 cells (AID-WT and AID-KO) that are sensitive (Ctrl) or resistant to Ibrutinib (blue) or idelalisib (red) treatment. Whiskers extend to a maximum of 1.5× interquartile range beyond the box. *p*-values are calculated by paired samples with a two-tailed Student’s *t*-test (* *p* < 0.05).

## Data Availability

Whole Exome Sequencing data are publicly available on https://www.ncbi.nlm.nih.gov/sra/, accessed on 2 April 2025) under the reference PRJNA1245283.

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
