# Peer review of "Evaluation of the Role of AID-Induced Mutagenesis in Resistance to B-Cell Receptor Pathway Inhibitors in Chronic Lymphocytic Leukemia"

_cimb, 2025, doi:10.3390/cimb47121031_

Round 1

Reviewer 1 Report

Comments and Suggestions for Authors

Results:

The author extracted genomic DNA from patients (treated/untreated). However, I didn't find any data regarding genomic analysis.

The author collected peripheral blood samples from patients, but didn't analyze their gene expression or Western blot. The author should justify their use of a sample for the experimentations. 

Discussion:

The author discusses their observations very briefly, and they didn't correlate with the observed data. The author should elaborate on the discussions as per the obtained results and their relevance to the literature and reported findings.

Material and methods:

The author extracted DNA from patients (before therapy and after therapy). Untreated: 10,  idelalisib: 8, and  ibrutinib:11 (Total 29), However author mentioned a total 56 samples.  The author should justify this ambiguity. 

The author included 10 Untreated patients, but the author should also include healthy volunteers as a control. 

in 4.3 section, line no. 250, " 5.5 × 106 HEK293FT". The author should check the number of cells or their notations.

To generate a resistant clone, the author treated the cells for 6 to 9 months. How did the author decide this time frame? The author should mention their reference to justify this dose duration.

Author Response

Thank you 

Reviewer 2 Report

Comments and Suggestions for Authors

Dear Authors,

The article “Evaluation of the Role of AID-Induced Mutagenesis in Resistance to B-Cell Receptor Pathway Inhibitors in Chronic Lymphocytic Leukemia» focuses on analyzing how mutations in PI3Kδ contribute to therapy resistance to idelalisib or ibrutinib since both drugs block PI3Kδ. The authors declare that AID-Induced mutations in the targeted genes do not make significant contribution, but there were mutations in off-target genes BCL2, MYC, and IRF8. This is a very important finding. The data were consistent across both human blood samples and cell lines. However, the overall narrative of the study would benefit from incorporating specific background details and emphasizing the key aspects. There are some comments below.

  1. A more precise introduction is needed. For example, was the contribution of the mutations of the PI3Kδ to the resistance caused by other drugs which have similar mechanism of action as idelalisib and ibrutinib? Are the any confirmed AID-induced mutations which lead to the resistance? What are the main known mechanisms of resistance to these drugs? MDR (P-glycoprotein) genes are not mentioned.
  2. The introduction would benefit from a sharper focus on the pre-existing knowledge about AID-induced mutations. It is important to clearly state what was not known in this field.
  3. Accordingly, the discussion should be expanded to include other potential factors that may contribute to idelalisib and ibrutinib resistance.
  4. Conclusion must include a statement on the future research directions needed to elucidate the factors of the resistance. The authors conclude that genomic instability could be a biomarker. Please, the authors showed that PI3Kδ blockade increases genomic instability in B cells (Introduction section: references [9,10], Figure 1A). Please clarify how the instability of the target genes can serve as a biomarker, and a biomarker for what specifically?
  5. I recommend moving Table 1 from the supplement to the main Results section. I experienced a technical difficulty accessing Supplementary Table 1. The table is critical for validating the patient cohort, as it is currently unclear how the development of treatment resistance was confirmed in all patients included in the study
  6. Please, clarify the development and characterization of the drug-resistant cell lines. Please specify the following:

- the fold-reduction in sensitivity (e.g., the increase in IC50 values) for each drug in the resistant sublines compared to their parental counterparts.

- any notable differences in degree of resistance between the two drugs.

Including these data in the Results section would be of significant interest to readers.

Author Response

Thank you

Round 2

Reviewer 2 Report

Comments and Suggestions for Authors

Dear Authors,

The article has been significantly improved, all my comments have been carefully taken into account in the corrected version.